# XIAP controls RIPK2 signaling by preventing its deposition in speck-like structures

Kornelia Ellwanger[1], Selina Briese[1], Christine Arnold[1], Ioannis Kienes[1], Valentin Heim[2,3], Ueli Nachbur[2,3], Thomas A Kufer[1]

The receptor interacting serine/threonine kinase 2 (RIPK2) is essential for linking activation of the pattern recognition receptors NOD1 and NOD2 to cellular signaling events. Recently, it was shown that RIPK2 can form higher order molecular structures in vitro. Here, we demonstrate that RIPK2 forms detergent insoluble complexes in the cytosol of host cells upon infection with invasive enteropathogenic bacteria. Formation of these structures occurred after NF-κB activation and depended on the caspase activation and recruitment domain of NOD1 or NOD2. Complex formation upon activation required RIPK2 autophosphorylation at Y474 and was influenced by phosphorylation at S176. We found that the E3 ligase X-linked inhibitor of apoptosis (XIAP) counteracts complex formation of RIPK2, accordingly mutation of the XIAP ubiquitylation sites in RIPK2 enhanced complex formation. Taken together, our work reveals novel roles of XIAP in the regulation of RIPK2 and expands our knowledge on the function of RIPK2 posttranslational modifications in NOD1/2 signaling.

## Introduction

The first line of defense in innate immunity in mammals encompasses several groups of pattern recognition receptors, including TLRs, cytosolic retinoic acid-inducible gene-I (RIG-I)-like helicases, and NOD-like receptors (NLRs), recognizing pathogen-associated molecular patterns (Janeway, 1989; Dostert et al, 2008). Most human NLRs are involved in innate and adaptive immunity via transcriptional regulation of MHC class I and class II or regulating the innate immune response (Ting et al, 2008). The NLR proteins NOD1 and NOD2 are intracellular pattern-recognition receptors, sensing bacterial peptidoglycan–derived γ-D-glutamyl-meso-diaminopimelic acid and muramyl dipeptide (MurNAc-L-Ala-D-isoGln, MDP), respectively (Girardin, Boneca et al, 2003a, 2003b; Chamaillard et al, 2003; Inohara et al, 2003). Activation of both, NOD1 and NOD2 induces the NF-κB pathway (Girardin et al, 2001; Ogura et al, 2001) by recruitment of the adaptor protein receptor interacting serine/threonine kinase 2 (RIPK2) (Inohara et al, 2000; Girardin et al, 2001). RIPK2 (RIP2/RICK/CARDIAK) belongs to the RIPK family, a group of serine/threonine protein kinases (Thome et al, 1998; Navas et al, 1999). However, RIPK2 lacks the RHIM domain found in the cell death associated members RIPK1 and RIPK3 (Humphries et al, 2015). RIPK2 is essential for NOD1-and NOD2-mediated NF-κB activation and might contribute to T-cell activation, whereas the latter point is controversial (Ruefli-Brasse et al, 2004; Hall et al, 2008; Tigno-Aranjuez et al, 2014; Nachbur et al, 2015).

The interaction of NOD1 and NOD2 with RIPK2 is mediated by heterotypic CARD–CARD interactions, involving residues in the exposed surfaces of the caspase activation and recruitment (CARD) domains of RIPK2 and NOD1/2 (Maharana et al, 2017; Manon et al, 2007; Mayle et al, 2014). In vitro, this can result in stable rope-like structures, which recently were proposed to be platforms for subsequent NF-κB activation (Gong et al, 2018; Pellegrini et al, 2018).

RIPK2 is controlled by complex posttranslational modification events, including autophosphorylation at several sites (Dorsch et al, 2006; Tigno-Aranjuez et al, 2010; Pellegrini et al, 2017). Best described are the phosphorylation events at S176 and Y474, which are associated with activity and structural changes (Pellegrini et al, 2017). The role and outcome of these phosphorylation events is not entirely understood. On the one hand, it was shown that kinase activity of RIPK2 is dispensable for signaling and might only affect protein stability (Abbott et al, 2004; Windheim et al, 2007). On the other hand, in addition to resulting in protein instability, inhibition of kinase activity by the tyrosine kinase inhibitors gefitinib and erlotinib or the RIPK2-specific compounds WEHI-345 and GSK583 was shown to reduce signaling (Tigno-Aranjuez et al, 2010; Nachbur et al, 2015; Haile et al, 2016). Some insight into this controversy was provided by the recent identification that RIPK2 inhibitors can also block interaction of RIPK2 with the E3 ubiquitin ligase X-linked inhibitor of apoptosis (XIAP), which is essential for RIPK2-mediated NF-κB activation (Goncharov et al, 2018). RIPK2 is modified by K63-, K27- and M1-linked ubiquitination at K209, located in its kinase domain (Hasegawa et al, 2008; Panda & Gekara, 2018). XIAP is the essential E3 for RIPK2 ubiquitination and interacts with RIPK2

[1]Department of Immunology, Institute of Nutritional Medicine, University of Hohenheim, Stuttgart, Germany [2]Walter and Eliza Hall Institute of Medical Research, Victoria, Australia [3]Department of Medical Biology, University of Melbourne, Melbourne, Australia

Correspondence: k.ellwanger@uni-hohenheim.de

through its baculoviral IAP-repeat (BIR) 2 domain (Krieg et al, 2009). XIAP also ubiquitinates K410 and K538 with K63-linked ubiquitin, which was shown to be important for NOD2 signaling (Goncharov et al, 2018). XIAP binding to RIPK2 recruits the linear ubiquitin chain assembly complex (LUBAC) (Damgaard et al, 2012). Moreover, further E3 ubiquitin ligases, including cellular inhibitor of apoptosis 1 (cIAP1) and cIAP2 (Bertrand et al, 2009), TNF receptor–associated factor (TRAF) 2 and TRAF5 (Hasegawa et al, 2008), and ITCH (Tao et al, 2009), were shown to participate in RIPK2 ubiquitination. However, their physiological roles remain to be clarified. Ubiquitination of RIPK2 leads to recruitment of the transforming growth factor β-activated kinase 1 (TAK1). This ultimately triggers the activation of the IκB kinase complex (Hasegawa et al, 2008) and MAPK signaling (Girardin et al, 2001).

Here, we provide novel insights into RIPK2 biology. We found RIPK2 to form high molecular weight complexes (RIPosomes) in the cytosol of epithelial cells upon infection with invasive bacterial pathogens such as *Shigella flexneri* and enteropathogenic *Escherichia coli*. Complex formation occurred after NF-κB activation, which was dependent on the CARD of NOD1 or NOD2, and autophosphorylation of RIPK2 at Y474. Inhibition of XIAP or the XIAP-mediated ubiquitination of RIPK2 induced sterile RIPosome formation, suggesting a key role of XIAP to prevent the deposition of RIPK2 in RIPosomes.

## Results

### RIPK2 forms cytosolic RIPosomes upon bacterial infection

To study details of RIPK2 function, we generated a stable inducible HeLa cell line expressing human EGFP-RIPK2. To activate RIPK2, we used the invasive Gram-negative bacterium *S. flexneri* that physiologically activates the NOD1 pathway (Girardin et al, 2001). EGFP-RIPK2 showed a cytoplasmic localization in mock infected cells and in cells infected with noninvasive *S. flexneri* BS176 but formed complexes (RIPosomes) within the cytosol in cells infected with the invasive *S. flexneri* M90T (Fig 1A). Infection with *S. flexneri* M90T increased IL-8 secretion compared with cells exposed to noninvasive *S. flexneri* BS176 or EGFP expressing control cell lines (Fig 1B). Albeit overexpression of RIPK2 led to some NF-κB activation and induction of IL-8 (McCarthy et al, 1998), the inflammatory response 6 h post infection (p.i.) was much higher, also in cells induced for EGFP-RIPK2 expression for 16 h, compared with noninduced cells (Fig 1B).

Immunoblotting of protein extracts of the induced EGFP-RIPK2 cells revealed an upshift of EGFP-RIPK2, starting 2 h p.i., and a concomitant decrease in phosphorylation of RIPK2 at S176, whereas NF-κB activation, measured by degradation of IκBα, seemed to start at earlier time points (Fig 1C). We found EGFP-RIPK2 to be enriched in the Triton X-100–insoluble pellet fraction at later times of infection, whereas EGFP-RIPK2 decreased in the Triton X-100–soluble fraction upon infection (Fig 1D). Similar results were obtained for endogenous RIPK2 from HeLa cells infected with *S. flexneri*. Endogenous RIPK2 decreased in the soluble fraction upon infection, whereas it accumulated in the pellet fraction. Although total RIPK2

levels slightly decreased with infection, an increase of the ratio of the RIPK2 pellet/total fraction at later time points of infection was evident (Fig 1E), showing that the HeLa EGFP-RIPK2 line reflects the properties of the endogenous protein.

Evaluation of the subcellular localization of RIPK2 during infection showed that EGFP-RIPK2 localized to F-actin–rich bacterial entry sites at early times of infection (30 min to 1 h p.i.) (Figs 1F and S1A). However, starting from 2 h p.i., formation of EGFP-RIPK2–positive dot-like structures throughout the cell was observed. These structures increased in volume over time, whereas dispersed cytoplasmic localized RIPK2 disappeared (Figs 1F and S1C, Video 1). The appearance of these structures upon infection followed the same kinetic as the change in the electrophoresis migration behavior of RIPK2 (Fig 1C and F). Immunostaining of p65 confirmed that NF-κB translocation into the nucleus preceded RIPosome formation (Fig 1G and H). qPCR analysis of IL-8 transcription further substantiated that RIPosomes formed after NF-κB activation (Fig 1I). Staining of *S. flexneri* LPS revealed that RIPosomes did not colocalize with *S. flexneri* (Figs 1A and S1B). *S. flexneri* infection is known to induce apoptosis in epithelial cells (Carneiro et al, 2009; Lembo-Fazio et al, 2011). However, the kinetic of activation of caspase-3 upon bacterial infection was not different between EGFP-RIPK2 expressing HeLa cells and unmodified HeLa cells (Fig S2A and B). Furthermore, formation of RIPK2 complexes did not coincide with caspase-3 activation or cell death in single cells (Fig S2A and B), supporting that RIPK2 complex formation was not associated with cell death.

Formation of RIPK2 complexes not only was not limited to *Shigella* but also seen upon infection with enteropathogenic *E. coli* (EPEC E2348/69). Although EPEC are described as noninvasive pathogens, some bacteria can invade epithelial cells (Donnenberg et al, 1989) (Fig S3B). Infection of HeLa EGFP-RIPK2 cells with EPEC led to RIPosome formation starting from 1 h p.i., whereas EPEC ΔescV, lacking the central component of the type III secretion system (Dupont et al, 2016), failed to induce RIPosomes. Visualization of F-actin by Lifeact-Ruby showed pedestal formation at 2 h p.i., which was absent in cells infected with the EPEC ΔescV strain (Fig S3A and D). Similar to *S. flexneri* infection, the amount of RIPosome-positive cells after infection with EPEC increased at later time points of infection (Fig S3C) and was associated with changed migration of the EGFP-RIPK2 protein in SDS–PAGE (Fig S3E). These data indicate that RIPosome formation is not only limited to *S. flexneri* infection but also occurs during infection with other Gram-negative enteropathogenic bacteria that can activate NOD1.

To identify if RIPosomes were associated with distinct cellular structures, we next performed co-immunofluorescence, using a set of markers for cellular compartments and innate immune signaling platforms associated with RIPK2. However, RIPosomes did not colocalize with any of the following proteins: TRAF-interacting forkhead-associated protein A (TIFA), which was recently reported to be involved in cytoplasmic innate immune responses towards *Shigella* (Gaudet et al, 2017; Garcia-Weber et al, 2018), TRAF6 (McCarthy et al, 1998), survival of motor neuron (SMN) protein, Gemin3 (Todd et al, 2010), EEA1 of endosomes (Irving et al, 2014), LC3 from phagosomes (Homer et al, 2012), and apoptosis inducing factor (AIF) from mitochondria (Fig S4).

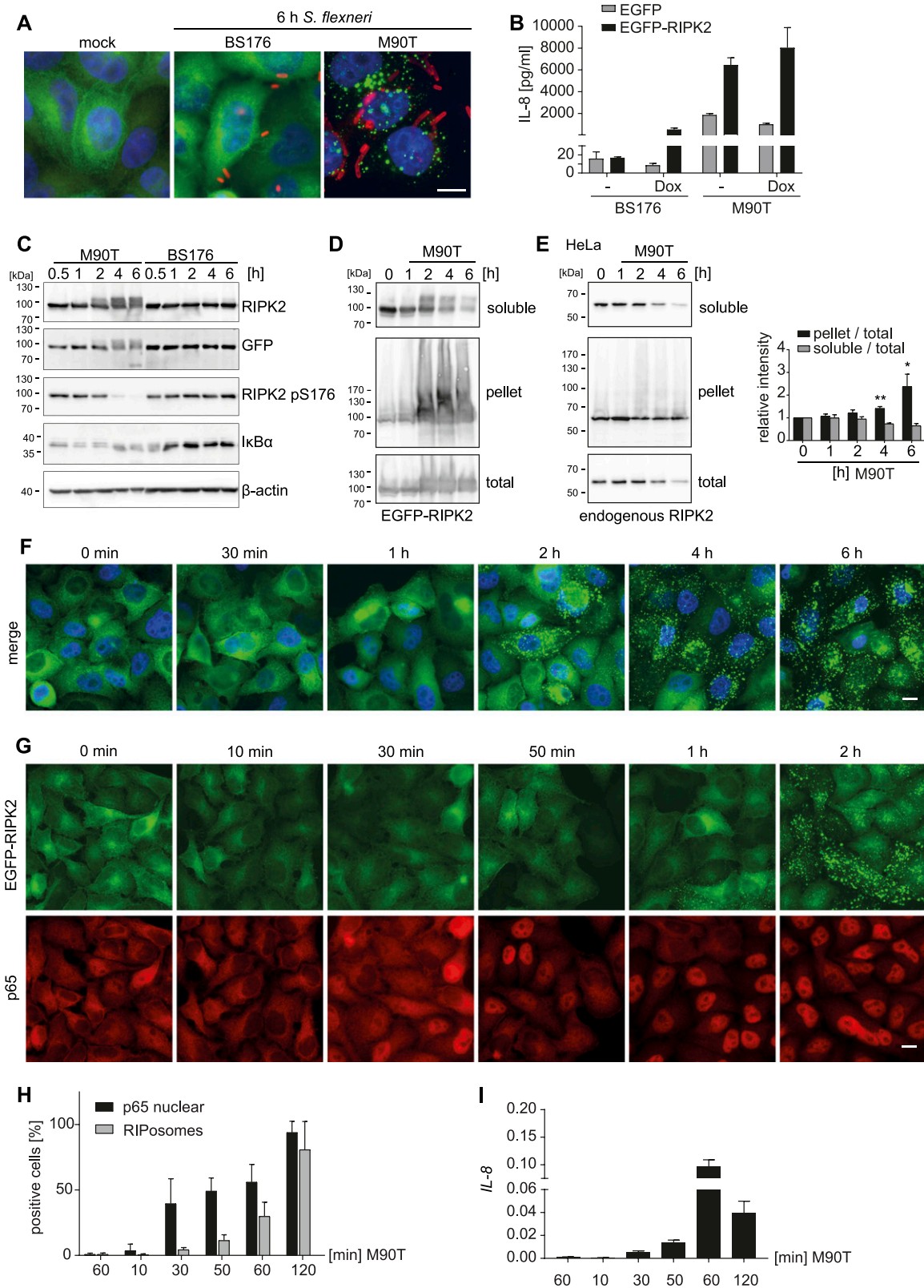

**Figure 1.   RIPK2 forms RIPosomes upon *Shigella* infection.**
**(A)** Indirect immunofluorescence micrographs of HeLa EGFP-RIPK2 cells infected with *S. flexneri* M90T or a noninvasive control strain (BS176) for the indicated time. Signal of anti–*Shigella*-LPS 5a antibody (red) and EGFP-RIPK2 (green) together with DNA staining (blue) is shown. Scale bar = 10 μm. **(B)** IL-8 secretion of doxycycline-induced versus noninduced HeLa EGFP-RIPK2 and HeLa EGFP cells 6 h p.i. with *S. flexneri* M90T or BS176. Mean + SD of one representative experiment conducted in

Taken together, we show that upon activation of epithelial cells by *S. flexneri*, RIPK2 forms distinct cytoplasmic structures. Complex formation followed NF-κB activation and was accompanied by a change in the electrophoretic migration behavior of RIPK2 and formation of triton insoluble aggregates for both endogenous and ectopically expressed RIPK2. RIPK2 complexes did neither co-localize with candidate cellular compartments, signaling platforms, nor with bacteria.

### RIPosome formation is dependent on the NOD1/2 CARD

To address whether RIPosome formation upon *Shigella* infection is dependent on NOD1, we performed siRNA-mediated knockdown of NOD1 before infection. 2 h p.i., a strongly reduced formation of RIPosomes was observed in NOD1 siRNA treated cells, compared with control siRNA (Fig 2A). Accordingly, immunoblot analysis showed an upshift of RIPK2 at 6 h p.i. in the control siRNA–treated sample, which was virtually absent in the siNOD1-treated sample (Fig 2B). As expected, NOD1 knockdown reduced IL-8 secretion upon infection but did not affect IL-8 secretion induced by autoactivation due to overexpression of RIPK2, compared with control siRNA-treated cells (Fig 2C). Activation of the canonical NF-κB pathway affects many cellular proteins and can lead to the activation of kinases, which might feedback on RIPK2. We, thus, tested the possibility that RIPosmes were dependent on canonical NF-κB activation. Treatment of the cells with TNF, a strong inducer of the canonical NF-κB pathway, failed to induce RIPosomes (Fig 2D). Moreover, RelA silencing did not change the kinetic of RIPosome formation upon bacterial infection as indicated by immunoblot and fluorescence microscopy (Fig 2E). Thus, *Shigella*-induced RIPosome formation is an event dependent on NOD1 but independent of canonical NF-κB activation. To analyze the prerequisites for RIPosome formation in greater detail, we used NOD1 and NOD2 with mutations in the NACHT domain, which we previously showed to affect NOD1/2 activation (Zurek et al, 2012). To this end, NOD1 K208R, D287A, E288A, and H517A and the NOD2 constructs K305R, E383A, and H603A along with NOD1 wt and NOD2 wt as well as the CARD domain of NOD1 alone, were transiently overexpressed in EGFP-RIPK2 cells. All constructs showed membrane localization dependent on the activity of the constructs as reported earlier (Zurek et al, 2012). Surprisingly, all tested NOD1 forms were capable to induce RIPosome formation, except for NOD1 ΔCARD (Fig 3A). Similarly, all NOD2 mutants were able to induce RIPosomes (Fig 3B). However, NOD1 and NOD2 were not recruited to RIPosomes in the final stage. Immunoblot analysis confirmed RIPosome formation by upshift of RIPK2 (Fig 3C), suggesting that the presence of an excess of NOD1/2 CARD is sufficient to trigger RIPosome formation.

Taken together, these results showed that RIPosome formation is dependent on the CARD of NOD1/2 but not on NF-κB activation. As RIPosomes can also be induced under sterile conditions by CARD overexpression, we can rule out that they depend on bacterial effector mechanisms.

### XIAP prevents RIPosome formation

Recently, it has been shown that loss of cellular inhibitor of apoptosis proteins (cIAPs) can lead to spontaneous RIPK1/3 Ripoptosome formation, following increased cell death (Feoktistova et al, 2011; Tenev et al, 2011). XIAP ubiquitinates RIPK2, a step essential for signaling downstream of NOD1/2 (Krieg et al, 2009). Disruption of the XIAP-RIPK2 interaction, thus, blocks RIPK2 ubiquitination and NF-κB signaling (Goncharov et al, 2018). Immunostaining of endogenous XIAP in HeLa EGFP-RIPK2 cells showed that XIAP was equally distributed throughout the cytoplasm in uninfected cells but co-localized to RIPosomes at 2 h p.i. (Fig 4A). siRNA-mediated targeting of XIAP, revealed that knockdown of XIAP was sufficient to induce RIPosome formation, independent of *Shigella* infection (Fig 4B). Similar results were obtained with three different siRNAs targeting XIAP (Fig S5A and B). Immunoblot analysis confirmed upshift of RIPK2 starting 2 h p.i. in cells treated with a control siRNA and infection independent upshift in cells with reduced XIAP levels (Fig 4C). We noticed that XIAP levels also declined upon bacterial infection and that this coincided with RIPK2 upshift, suggesting that XIAP activity and protein abundance negatively regulate RIPosome formation (Fig 4C). In agreement with previous reports, IL-8 secretion was completely abrogated after XIAP knockdown (Fig 4D). Finally, to validate the results obtained by siRNA, we overexpressed second mitochondrial-derived activator of caspases (SMAC), which antagonize XIAP function (Du et al, 2000). To this end, ubiquitin fusion constructs, which expose their AVPI motive upon processing in the cytosol, were used (Kashkar et al, 2006). Both full-length and a truncated construct, missing the mitochondrial localization site, reduced the level of XIAP and induced RIPosome formation, whereas overexpression of the construct missing the AVPI site, which is essential for XIAP antagonization, did not induce RIPosomes nor upshift of RIPK2 in the immunoblot (Fig S5C and D). To corroborate our findings in other cell types, we used human myeloid THP-1 cells and found that stimulation with the NOD2 agonist L18-MDP partly shifted endogenous RIPK2 into the Triton X-100 insoluble fraction (Fig 4E). Combined treatment of THP-1 cells with L18-MDP and the SMAC mimetic compound A led to the change of endogenous RIPK2 into the Triton X-100 insoluble fraction (Fig 4E).

XIAP is an ubiquitin ligase, and several sites in RIPK2 have been shown to be ubiquitinated by XIAP (Witt & Vucic, 2017). The most

triplicates is shown. **(C)** Immunoblot analysis of HeLa EGFP-RIPK2 cells infected with *S. flexneri* M90T or BS176 for the indicated time. **(D)** Whole cell Triton X-100 lysates of HeLa EGFP-RIPK2 cells infected with *S. flexneri* M90T for the indicated time were separated into soluble and insoluble fractions and analyzed by immunoblot using anti-RIPK2 antibody. **(E)** Whole cell Triton X-100 lysates of HeLa cells infected with *S. flexneri* M90T for the indicated time were separated into soluble and insoluble fractions and analyzed by immunoblot using anti-RIPK2 antibody. Right panel: relative intensity of pellet to total fraction and soluble to total fraction (n = 3). All data are representative of at least two independent experiments. *P < 0.05, **P < 0.005 (*t* test). **(F)** Fluorescence micrographs of HeLa EGFP-RIPK2 cells infected with *S. flexneri* M90T for the indicated time. EGFP-RIPK2 signal (green) merged with DNA staining (blue) is shown. Scale bar = 10 *µ*m. **(G)** Indirect immunofluorescence micrographs of HeLa EGFP-RIPK2 cells infected for the indicated time with *S. flexneri* M90T. Signals for EGFP-RIPK2 (green) and p65 (red) are shown. Scale bar = 10 *µ*m. **(H)** Quantification of RIPosomes and nuclear p65 staining in the cells from (G). For each time point more than 200 cells were quantified. Mean + SD of two independent experiments is shown. **(I)** RT-qPCR of *IL-8* mRNA expression in HeLa EGFP-RIPK2 cells at early time points of infection SD of triplicate measurements of one representative experiment is shown.

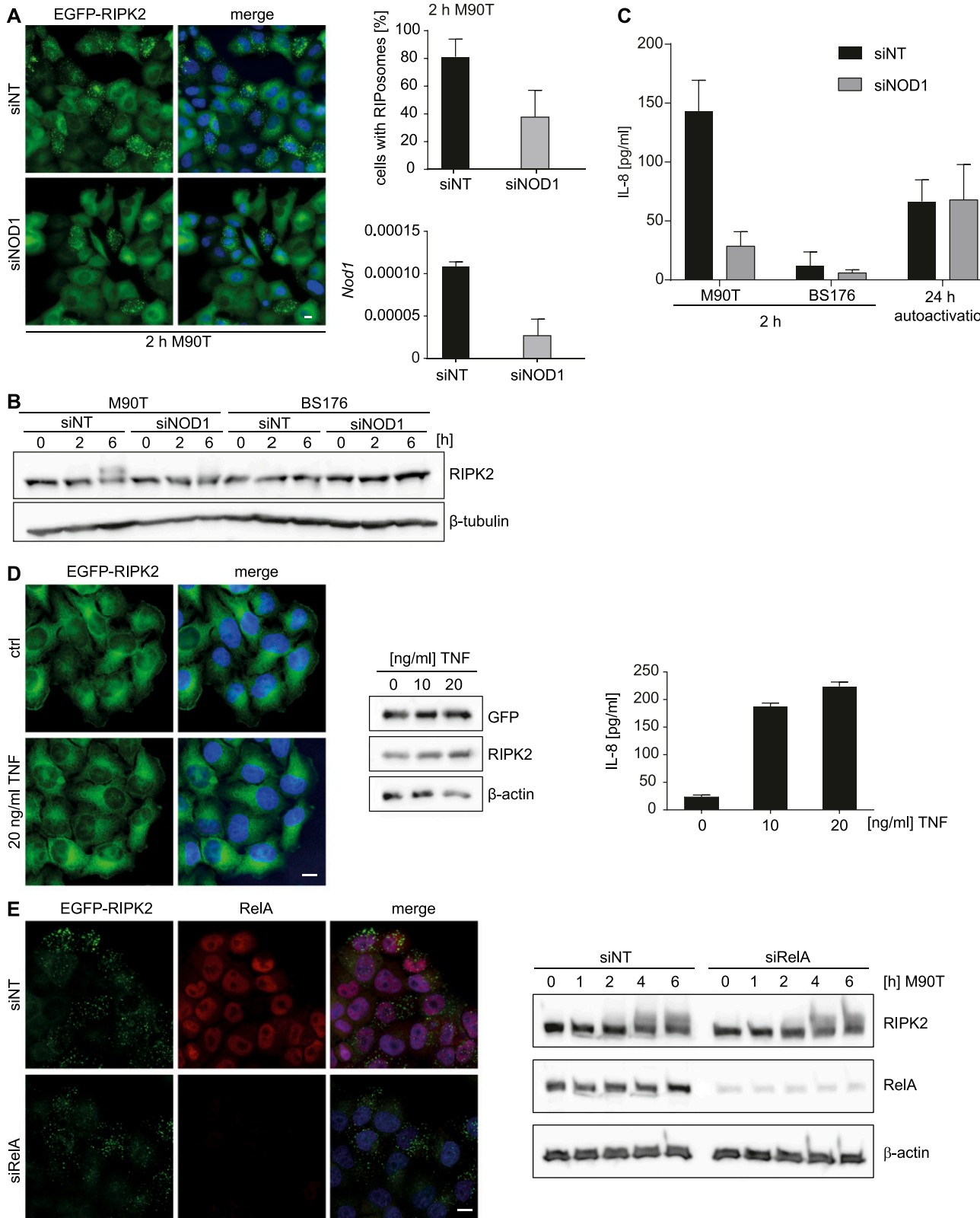

**Figure 2. *Shigella*-induced RIPosome formation depends on NOD1 but not NF-κB activation.**
**(A)** Left panel: fluorescence micrographs of HeLa EGFP-RIPK2 cells, treated for 72 h with *NOD1* siRNA or a nontargeting siRNA, after 2-h *S. flexneri* M90T infection. EGFP-RIPK2 (green) and merge with DNA staining (blue) are shown. Scale bar = 10 μm. Upper right panel: quantification of RIPosomes in the cells. 500 cells of n = 2 experiments were quantified. Lower right panel: RT-qPCR analysis of *NOD1* mRNA expression 72 h after siRNA transfection. SD of triplicate measurements of one representative

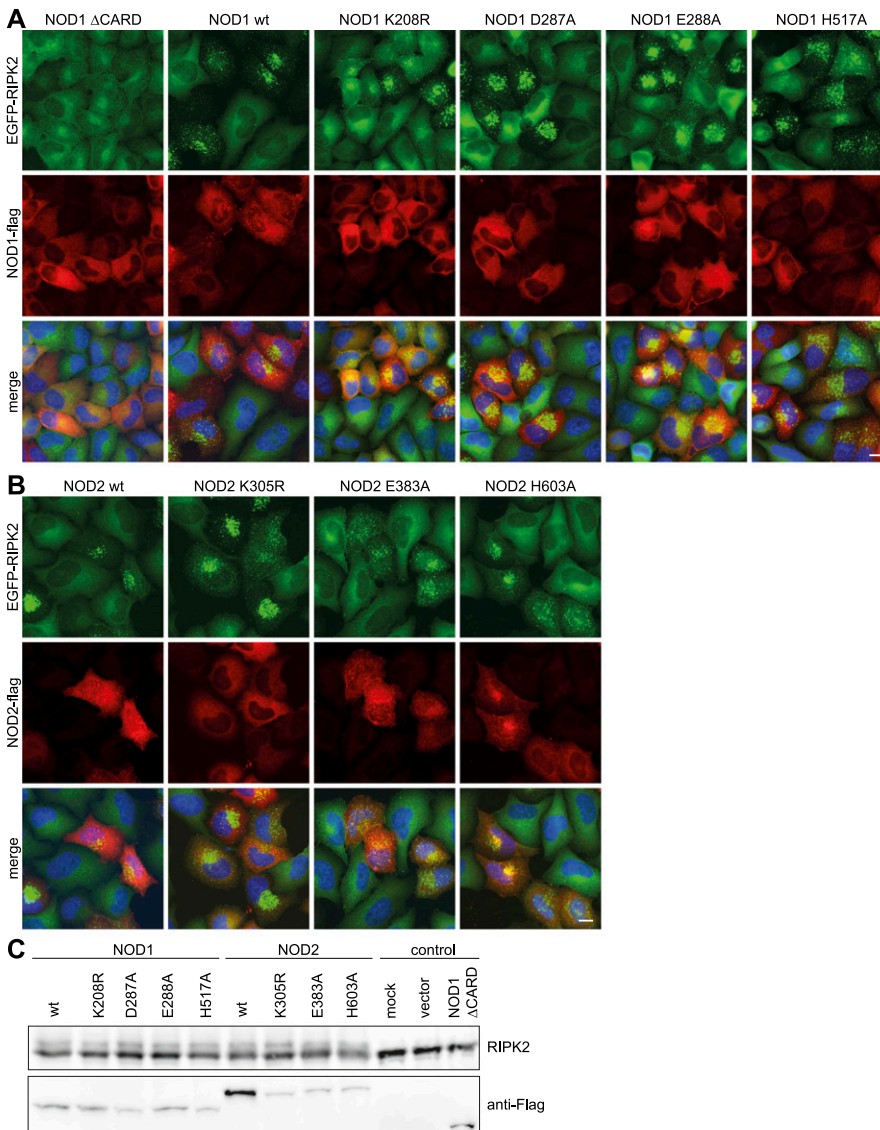

**Figure 3.  RIPosome formation is dependent on the CARD of NOD1/2.**
**(A, B)** Indirect immunofluorescence micrographs of HeLa EGFP-RIPK2 cells transfected with (A) NOD1-Flag or NOD1-Flag mutants or (B) NOD2-Flag and or NOD2-Flag mutants. Signals of EGFP-RIPK2 (green), NOD1/2-Flag (red), and merge with DNA staining (blue) are shown. Scale bar = 10 μm. **(C)** Immunoblot analysis of the cells in (A) and (B), probing for RIPK2, Flag, and β-actin as loading control.

prominent are K209 (Hasegawa et al, 2008; Panda & Gekara, 2018), K410, and K538 (Goncharov et al, 2018). As XIAP depletion induced RIPosome formation in our cell line, we suggest that this is due to diminished ubiquitination of RIPK2. To test this hypothesis, we generated a stable cell line expressing RIPK2 K209/410/538R (referred to here as RIPK2 3KR). Analysis of this line compared with the EGFP-RIPK2 cells showed that the RIPK2 3KR cells spontaneously formed RIPosomes, as shown by immunofluorescence (Fig 5A and B) and showed upshifted electromobility of RIPK2 (Fig 5C). By contrast, these mutations blocked *Shigella*-induced IL-8 secretion (Fig 5D) and degradation of XIAP upon infection (Fig 5C). This provides strong indirect evidence that XIAP-mediated ubiquitination of RIPK2 at K209, K410, and K538 is required for signaling and to block deposition of RIPK2 in RIPosomes.

experiment is shown. **(B)** Immunoblot analysis of HeLa EGFP-RIPK2 cells, treated with a *NOD1* specific or a nontargeting siRNA. The cells were infected 72 h after siRNA treatment with *S. flexneri* M90T or BS176 for the indicated time. Protein levels were detected using anti-RIPK2, and anti–β-tubulin as loading control. **(C)** IL-8 levels in supernatants from (A) in comparison with supernatants taken 24 h after doxycycline treatment with 1 μg/ml (autoactivation). Mean of n = 2 experiments with SD is shown. **(D)** Left panel: fluorescence micrographs of HeLa EGFP-RIPK2 cells treated for 18 h with TNF. EGFP-RIPK2 (green) and merge with DNA staining (blue) are shown. Right panel: immunoblot for GFP-RIPK2 detected by anti-GFP and anti-RIPK2 antibodies in cells treated with different amounts of TNF for 6 h. IL-8 release in the supernatant of these cells is shown in the graph at the right. Mean + SD of triplicate measurements of one representative experiment is shown. **(E)** HeLa EGFP-RIPK2 cells infected with *S. flexneri* M90T. Left panel: fluorescence micrographs showing GFP-RIPK2 (green) and RelA staining (red) and merge with DNA staining (blue) at 2 h p.i. Right panel: immunoblot analysis of cells at different time points p.i.

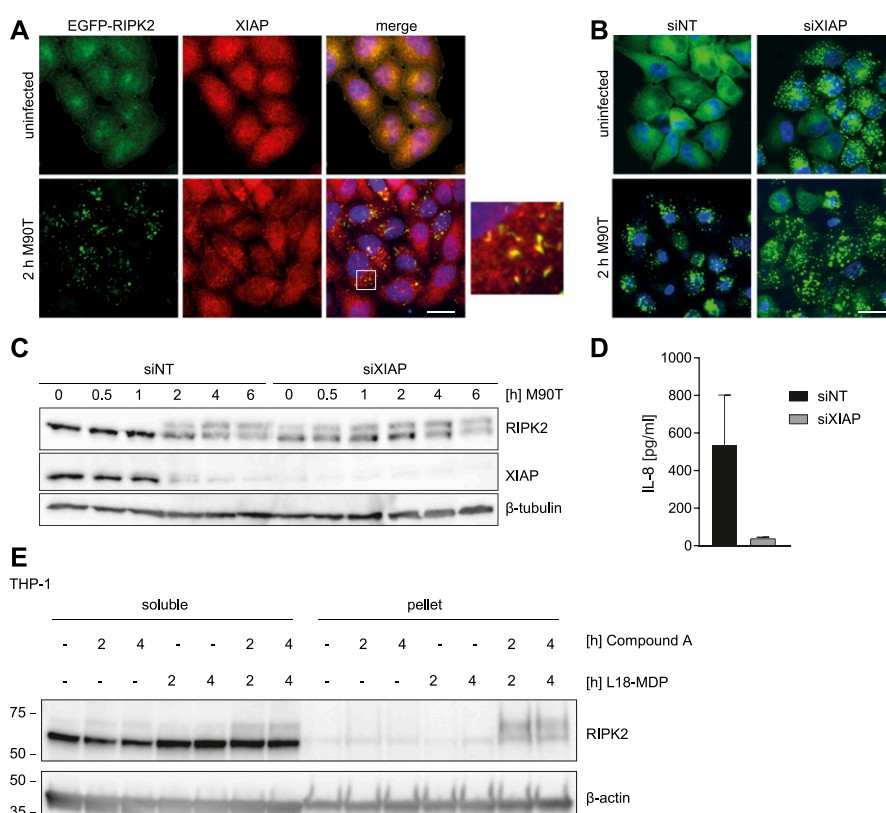

**Figure 4. XIAP prevents RIPosome formation.**
**(A)** Indirect immunofluorescence micrographs of HeLa EGFP-RIPK2 cells infected for 2 h with *S. flexneri* M90T. Signals for EGFP-RIPK2 (green), XIAP (red), and merge with DNA staining (blue) are shown. Co-localization is shown at higher magnification in the inlay. Scale bar = 10 *μ*m, box: 6.6 × 6.6 *μ*m. **(B)** Fluorescence micrographs of HeLa EGFP-RIPK2 cells, treated for 48 h with *XIAP* siRNA or a nontargeting siRNA, after infection with *S. flexneri* M90T for 2 h or left uninfected. EGFP-RIPK2 and merge with DNA staining are shown. Scale bar = 10 *μ*m. **(C)** Immunoblot analysis of HeLa EGFP-RIPK2 cells treated for 48 h with *XIAP* siRNA or a nontargeting siRNA and infected with *S. flexneri* M90T for the indicated time. Immunoblot was probed with anti-RIPK2 antibody, anti-XIAP, and anti-*β*-tubulin as loading control. **(D)** IL-8 levels in the supernatants from HeLa EGFP-RIPK2 cells, treated for 48 h with an *XIAP* siRNA or a nontargeting siRNA, after infection with *S. flexneri* M90T for 6 h. Mean of n = 2 with SD is shown. **(E)** Immunoblot analysis of endogenous RIPK2 from THP-1 cells upon stimulation with 1 *μ*g/ml Compound A and 200 ng/ml L18-MDP for the indicated times. Whole cell Triton X-100 lysates were separated into soluble and insoluble (pellet) fractions and analyzed by immunoblot using anti-RIPK2 antibody.

## Phosphorylation of RIPK2 at Y474 is essential for RIPosome formation

Several autophosphorylation sites of RIPK2 have been reported, S176 and Y474 being the best studied. S176 is described as autophosphorylation site important for RIPK2 catalytic activity (Dorsch et al, 2006), whereas autophosphorylation at Y474 is essential for full NOD signaling (Tigno-Aranjuez et al, 2010). We hypothesized that these phosphorylation sites might have a role in RIPosome formation, as phosphorylation at S176 declined upon infection and subsequent RIPosome formation (Fig 1C), whereas tyrosine phosphorylation appeared simultaneously (Fig S6A). Transient over-expression of EGFP-RIPK2 S176A, EGFP-RIPK2 S176E, and EGFP-RIPK2 Y474F in HeLa cells showed that both RIPK2 S176A and RIPK2 S176E induced RIPosomes similar to WT RIPK2, whereas RIPK2 Y474F was not able to form complexes, as shown by immunofluorescence and immunoblot (Fig S6B and C). In HEK293T cells, expression of increasing amounts of S176A and S176E, but not RIPK2 Y474F, induced NF-κB activation at least as high as RIPK2 wt (Fig S6D). Stable cell lines expressing inducible EGFP-RIPK2 S176A, EGFP-RIPK2 S176E, and EGFP-RIPK2 Y474F mutants confirmed that RIPK2 Y474F was not able to form RIPosomes (Fig 5E and F). Using these cell lines with controlled expression of RIPK2, we observed differences in the kinetics of RIPosome formation for the S176 mutants. S176E showed RIPosome formation similar to WT RIPK2 protein, whereas RIPK2 S176A led to RIPosome formation and upshift earlier upon infection, as compared with wt (Fig 5F and G). Accordingly, RIPK2 S176A

induced more, and RIPK2 S176E less, IL-8 mRNA expression at early time points of infection, compared to wt RIPK2 (Fig 5H and I). By contrast, mutation of Y474 resulted in an RIPK2 protein incapable to induce neither IL-8 nor RIPosomes upon bacterial infection (Fig 5H and I). In line with lack of RIPosome formation by RIPK2 Y474F, this protein remained in the soluble fraction upon bacterial infection of the cells (Fig S6E). Expression of EGFP-RIPK2 and EGFP-RIPK2 Y474F in both human and mouse myeloid cells (THP-1 and Raw 264.7) validated RIPosome formation for RIPK2 and confirmed a lack of both upshift and RIPosome formation for the Y474F protein (Fig S6F). Notably, we observed that XIAP levels were reduced upon infection in cells expressing RIPK2, RIPK2 S176A, and RIPK2 S176E but not in cells expressing RIPK2 Y747F and correlated with the induction of RIPosome formation (Fig 5G). As XIAP depletion induced sterile RIPosome formation, this suggests that XIAP might be central in controlling RIPosome formation also upon bacterial infection.

Inhibition of RIPK2 with the tyrosine kinase inhibitor gefitinib (Tigno-Aranjuez et al, 2010, 2014) and the RIPK2-specific compound GSK583 (Haile et al, 2016) also led to spontaneous induction of RIPosomes independent of infection (Fig S7A). Notably, the appearance of the formed complexes was different for the two compounds: gefitinib led to the formation of dot-like structures, whereas fiber-like structures were obtained with GSK583 (Fig S7A). RIPK2 complex formation was accompanied by appearance of higher molecular weight signals for both inhibitors at later time points of infection. Interestingly, XIAP levels were preserved and S176 phosphorylation increased in the course of *S. flexneri* M90T

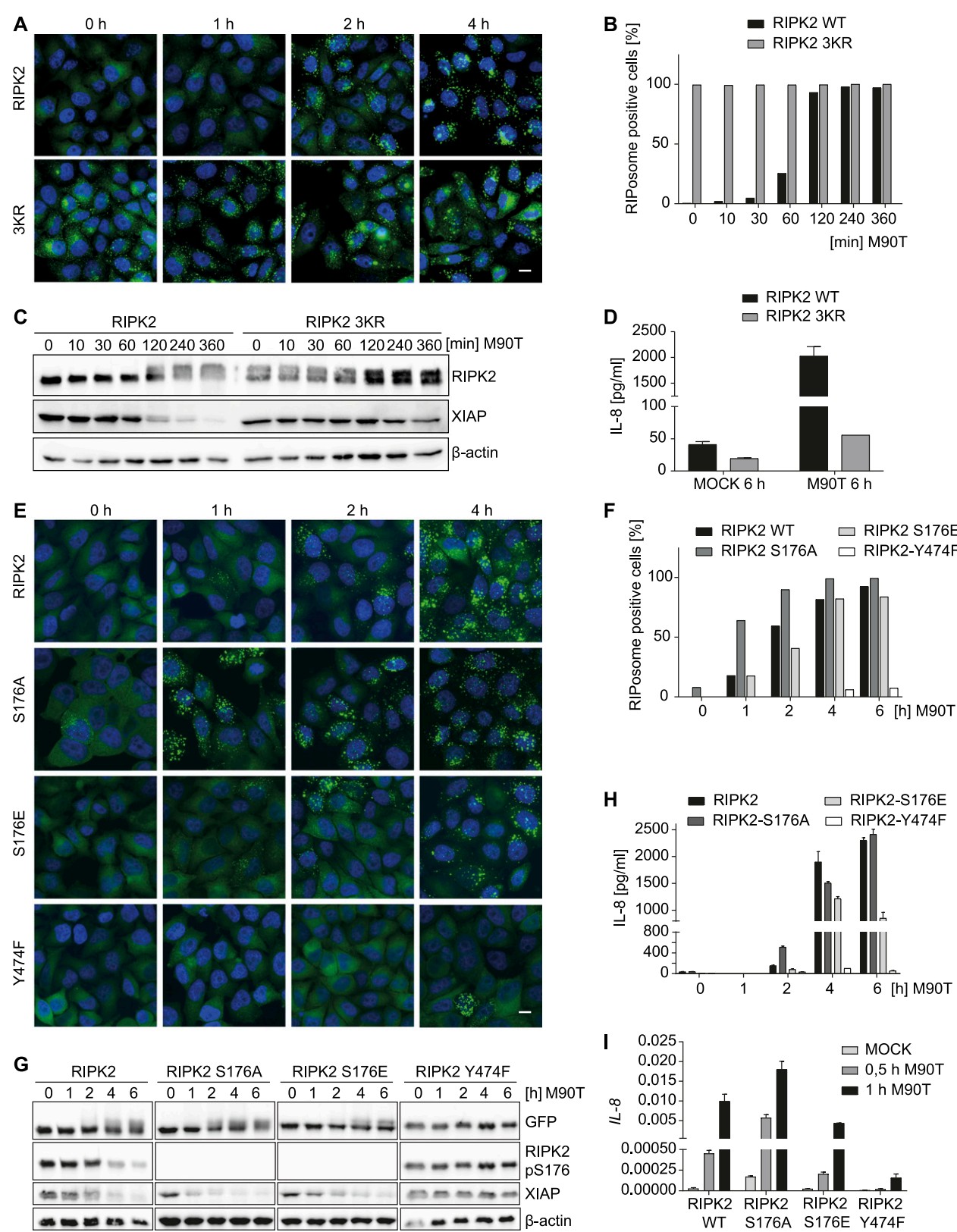

**Figure 5. Ubiquitination and phosphorylation of RIPK2 are involved in the regulation of RIPosome formation.**
**(A)** Fluorescence micrographs of stable HeLa lines expressing EGFP-RIPK2 or EGFP-RIPK2 3KR (K209R, K410R, and K538R) infected for the indicated time with *S. flexneri* M90T. EGFP (green) and merge with DNA staining (blue) are shown. Scale bar = 10 μm. **(B)** Quantification of RIPosome containing cells of (A) in the course of *S. flexneri* M90T infection. At least 200 cells per time point were counted. **(C)** Immunoblot analysis of EGFP-RIPK2 and EGFP-RIPK2 3KR cells infected for the indicated time with *S. flexneri*

infection when cells were pretreated with GSK583, whereas gefitinib led to reduced XIAP and pS176 levels (Fig S7B). Both substances inhibited *Shigella*-induced IL-8 responses, whereby GSK583 led to stronger reduction of IL-8 levels compared with gefitinib (Fig S7C).

Finally, we tested if inhibition of proteasomal degradation had an influence on steady-state RIPosome formation in our HeLa EGFP-RIPK2 cell line. We observed that MG132 induced RIPosomes starting 4 h posttreatment as visualized by both immunofluorescence and immunoblot upshift of RIPK2 (Fig S7D).

In conclusion, these results demonstrate an essential role of phosphorylation of RIPK2 at Y474 for the inflammatory function and RIPosome formation, whereas lack of phosphorylation at S176 can facilitate complex formation.

# Discussion

Here, we provide novel insights into the molecular regulation of RIPK2 and show that upon bacterial invasion, RIPK2 forms high molecular weight complexes in the cytosol, which we and others termed RIPosomes (Gong et al, 2018). Formation of these structures was accompanied by reduced detergent solubility of RIPK2 and changes in the electrophoretic mobility. RIPK2 forms hetero-oligomers via CARD–CARD interactions and was found to polymerize to long filamentous structures in vitro, which were proposed to act as signaling platforms for the activation of pro-inflammatory signaling downstream of NOD1 and NOD2 (Gong et al, 2018; Pellegrini et al, 2018). Our data support that RIPosome initiation depends on availability of the CARD domain of NOD1/2 that serves as seed for aggregation. Although we ruled out that canonical NF-κB activation is needed for RIPosome formation, we cannot formally exclude that other signaling pathways or activation of kinases acting on RIPK2 might affect RIPosome formation. Recent in vitro structural data that indicate that the CARD of NOD1 forms a transient unstable complex with RIPK2 CARD to induce aggregation is consistent with our in situ data (Gong et al, 2018). NF-κB signaling thereby seems to be uncoupled from this process as mutations in NOD1 and NOD2 that interfere with NF-κB activation still triggered RIPosome formation. In addition, mutations in the CARD interaction surface of NOD1 and RIPK2 that interfere with signaling but not with protein interaction are described (Mayle et al, 2014). This suggests that distinct molecular surfaces are involved in signaling versus RIPosome formation. As the NOD1 CARD offers several possible interaction modes with RIPK2, these might define different biological outcomes (Maharana et al, 2017). Formation of higher molecular complexes is also known for other members of the RIPK family. RIPK1 and RIPK3 can form a complex called the Ripoptosome, a cell death platform, discriminating between apoptosis and necroptosis (Feoktistova et al, 2011). However, RIPK2 does not

possess the RHIM domain essential for cell death events and oligomerization in RIPK1 and RIPK3. In addition, our data show that formation of RIPosomes is not directly associated with apoptosis or cell death in HeLa cells.

We found that RIPosomes formed after NF-κB activation mediated by NOD1, which is contradictory to a direct role of these structures in NF-κB activation by NOD1/2, as recently proposed by others (Gong et al, 2018; Pellegrini et al, 2018). Activation of NF-κB responses by NOD1 and NOD2 is associated with membrane recruitment of NOD1 and NOD2 (Lecine et al, 2007; Kufer et al, 2008; Travassos et al, 2010; Nakamura et al, 2014). In line with the formation of RIPosomes after this initial signaling event, we did not observe recruitment of neither NOD1 nor NOD2 to RIPosomes. Moreover, knockdown and inactivation of XIAP, which is essential for NF-κB activation by RIPK2 (Krieg et al, 2009; Damgaard et al, 2013; Andree et al, 2014; Goncharov et al, 2018), induced RIPosomes in sterile conditions, again supporting that RIPosome formation is independent of inflammatory signaling by RIPK2. *Shigella* induces nonapoptotic SMAC release in epithelial cells to counteract NOD1-mediated sensing by inhibition of XIAP (Andree et al, 2014). This process might be crucial for RIPosome formation upon *Shigella* infection by inhibition of XIAP function. Ubiquitination plays a central role in controlling RIPKs (reviewed in Witt and Vucic (2017)). Notably, RIPK1/RIPK3 Ripoptosome formation is also induced by loss of XIAP (Tenev et al, 2011; Yabal et al, 2014). Mechanistically, this was proposed to involve aberrant ubiquitination of RIPK1/3, raising the question which ubiquitination events play a role in RIPK2 complex formation. As RIPK2 is mainly regulated by ubiquitination on K209 via covalent attachment of different linkage types of ubiquitin (Hasegawa et al, 2008; Panda & Gekara, 2018), this site might play a role in RIPosome formation as well. Our data show that, ubiquitination at three sites, including K209, is needed for activation of RIPK2 and led to RIPK2 deposition in RIPosomes when mutated/unubiquitinated. Although lack of ubiquitination at these sites apparently induced RIPosome formation, inhibiting proteasome degradation also led to induction of RIPosomes, suggesting that other K48-modified lysine(s) in RIPK2 contribute to this process. In our HeLa EGFP-RIKP2 cell line, we observe some autoactivation by overexpression of RIPK2; this might suggest that RIPosomes are compartments that are formed to degrade and, thus, limit RIPK2 signaling. Accordingly, we observed insoluble RIPK2 complexes in THP-1 cells upon activation of NOD2 by L18-MDP and simultaneous targeting of XIAP by SMAC mimetics. That SMAC mimetics alone did not induce RIPK2 modification in these cells likely is related to the fact that there is no basal RIPK2 activation in THP-1 cells compared with the situation in HeLa EGFP-RIPK2 cells, which showed some IL-8 response upon induction of EGFP-RIPK2 expression (Fig 1B). XIAP, thus, might act as a master switch between the NF-κB pathway and RIPosome signaling.

M90T. Probing for RIPK2, XIAP, and β-actin as loading control. **(D)** IL-8 levels in supernatants of EGFP-RIPK2 and EGFP-RIPK2 3KR cells infected for 6 h with *S. flexneri* M90T. Mean + SD from three biological replicates with two technical replicates each is shown. **(E)** Fluorescence micrographs of stable HeLa lines expressing EGFP-RIPK2, EGFP-RIPK2 S176A, EGFP-RIPK2 S176E, and EGFP-RIPK2 Y474F infected for the indicated time with *S. flexneri* M90T. EGFP (green) and merge with DNA staining (blue) are shown. Scale bar = 10 μm. **(F)** Quantification of RIPosome containing the cells of (A) in the course of *S. flexneri* M90T infection. At least 200 cells per time point were counted. **(G)** Immunoblot analysis of the cells from (A) infected for the indicated time with *S. flexneri* M90T. Probing for RIPK2, RIPK2 pS176, and β-actin as loading control. **(H)** IL-8 levels in supernatants of cells from (E) infected for 6 h with *S. flexneri* M90T. **(I)** RT-qPCR of *IL-8* mRNA expression in cells from (E) at early time points of infection.

Surprisingly, we found that the RIPK2 inhibitors gefitinib and GSK583 also induced spontaneous formation of RIPosomes. The quality of the formed structures, thereby, was different between the two inhibitors. GSK583 blocks XIAP binding to RIPK2 (Goncharov et al, 2018), which is consistent with our data on XIAP depletion and can explain RIPosome formation by XIAP targeting. However, gefitinib, a tyrosine kinase inhibitor does not interfere with the XIAP–RIPK2 interaction (Goncharov et al, 2018); how this compound induces RIPosome formation without XIAP targeting and in view that Y474 phosphorylation on RIPK2 is needed for RIPosome formation needs to be established in further work.

Besides ubiquitination, phosphorylation events play pivotal roles in regulating RIPK2 activity and function. Autophosphorylation of RIPK2 at several sites is associated with activation and dimer formation (Pellegrini et al, 2017). S176 is one of the best described sites that contribute to RIPK2 activation and functionality (Dorsch et al, 2006; Pellegrini et al, 2017). We observed that RIPosome formation coincided with loss of phosphorylation at this site. Accordingly, mutation of this site to alanine or phosphomimetic glutamic acid did not affect RIPosome formation per se. The S176A mutant showed higher autoactivation and favored complex formation, whereas the phosphomimetic mutant (S176E) showed decreased activity and delayed complex formation. These first sight contradicting results can be explained when assuming that loss of phosphorylation at S176 is a prerequisite for RIPK2 activation. By contrast, mutation of Y474 to phenylalanine in the CARD completely impaired RIPosome formation and signaling. This is in line with the observation that nonaggregated RIPK2 is present in a highly phosphorylated form (Pellegrini et al, 2018). Phosphorylation at the Y474 corresponding site in Apoptosis-associated speck-like protein containing a CARD (ASC) was associated with ASC speck formation (Hara et al, 2013), and it was proposed that this site in RIPK2 might have similar functions (Boyle et al, 2014). Our data support a role of this site in aggregate formation for RIPK2. Such phosphorylation patterns might be general regulators of CARD aggregate formation and signaling, as ASC speck formation is dependent on phosphorylation, whereas signaling and interaction of ASC with NLRP3 is not (Hara et al, 2013). In contrast to ASC specks, RIPosomes appeared as multiple high molecular weight complexes per cell and increase over time of infection, suggesting different mechanism of oligomerization of these CARD proteins. The abovementioned phosphorylation events in RIPK2 could also indirectly contribute to RIPosome formation. XIAP protein levels inversely correlated with RIPosome formation for all tested RIPK2 variants and reduction of XIAP levels by siRNA led to spontaneous RIPosome formation. These observations suggest a central role of XIAP in the RIPosome pathway.

Although ectopic expression of RIPK2 was used in some experiments, we provide ample evidence to show that the controlled expression of RIPK2 in our stable HeLa cell lines well reflects the endogenous situation. We show that endogenous RIPK2 forms detergent-insoluble complexes with similar kinetics in HeLa cells and that endogenous RIPK2 in myeloid cells also form detergent-insoluble RIPK2 upon NOD2 activation and XIAP targeting. The use of our EGFP-RIPK2 cell lines allowed the gain of further insights into RIPK2 activation, and we show that they are well-suited reporters for cytosolic bacterial recognition in real time.

To summarize, our data suggest that RIPosomes act to dampen RIPK2 signaling or might acts as signaling platform with yet to be identified functions.

# Materials and Methods

### Plasmids and reagents

The plasmids pcDNA5/FRT/TO-EGFP and pcDNA5/FRT/TO-EGFP-RIPK2 were generated by molecular cloning. Site-directed mutagenesis was used to generate RIPK2 S176A, RIPK2 S176E, RIPK2 Y474F, and RIPK2 K209/410/538R. Flag-tagged NOD1, NOD2, and the respective mutants are described in (Kufer et al, 2006; Kufer, 2008). pLifeact-Ruby was kindly provided by Roland Wedlich-Söldner (Riedl et al, 2008). Ubi-SMAC-fl, Ubi-SMAC-tr, and Ubi-SMAC-AVPI were kindly provided by Hamid Kashkar (University of Cologne) (Kashkar et al, 2006). pCR3-Flag-TRAF6 was kindly provided by Greta Guarda (IRB). pcDNA3_TIFA-myc was kindly provided by Jessica Yu (Huang et al, 2012). For transient overexpression, the plasmids were transfected, using Lipofectamine 2000 (Thermo Fisher Scientific). Reagents used were gefitinib and human TNFα (InvivoGene), GSK583 (Cayman Chemicals), MG132 (Sigma-Aldrich), Compound A (TetraLogic Pharmaceuticals), and L18-MDP (Bachem).

### Cells and cell culture

HeLa, HEK293T, and RAW 264.7 cells were cultured in DMEM supplemented with 10% heat-inactivated FBS and antibiotics (penicillin and streptomycin). THP-1 cells were cultured in RPMI supplemented with 10% heat-inactivated FBS and antibiotics. All lines were grown at 37°C with 5% $CO_2$ and continuously tested for the absence of mycoplasma by PCR.

Stable cell lines were generated by co-transfection of pcDNA5/FRT/TO and pOG44 in a 9:1 ratio into HeLa FlpIN T-REx cells (kindly provided by the Hentze Lab, EMBL) using Lipofectamine 2000 (Thermo Fisher Scientific) and selection with 10 µg/ml blasticidin and 500 µg/ml hygromycin B. The HeLa FlpIN T-REx EGFP-RIPK2 Lifeact-Ruby cell line (EGFP-RIPK2/Lifeact) was generated by transfection of pLifeact-Ruby and selection with 250 µg/ml G418. EGFP- or EGFP-RIPK2 expression was induced with 1 µg/ml doxycycline for 16 h before infection or live cell imaging if not stated otherwise. HeLa, HEK293T, RAW 264.7, and THP-1 cells were obtained from ATCC.

### Bacteria and bacterial infection

*S. flexneri* M90T *afaE* and BS176 *afaE* (Clerc & Sansonetti, 1987) were kindly provided by Philippe Sansonetti (Institute Pasteur) and were grown in Caso Broth containing 200 µg/ml spectinomycin. EPEC E2348/69 wt (streptomycin) and EPEC E2348/69 ΔescV (escV::min-iTn10*kan*, streptomycin, kanamycin) were kindly provided by Mathias Hornef (RWTH Aachen) and were grown in Caso Broth or LB supplemented with appropriate antibiotics (Dupont et al, 2016).

Infection with *Shigella* was performed at an multiplicity of infection of 10 in DMEM without supplements. After 15 min of bacterial

sedimentation at room temperature, infection was started at 37°C and 5% $CO_2$. After 30 min, the medium was changed to 250 $\mu$l DMEM containing 100 $\mu$g/ml gentamycin. For ELISA, the supernatants were collected at the indicated time p.i.

EPEC infection was performed in 24-well plates at an multiplicity of infection of 25 in a volume of 250 $\mu$l DMEM without supplements. EPEC were centrifuged onto the cells for 5 min at 560$g$, followed by infection at 37°C and 5% $CO_2$. For gentamycin protection assay, the cells were lysed with 1 ml 0.1% Triton X-100 in PBS, serial dilutions were prepared in PBS, streaked onto agar plates containing 200 $\mu$g/ml spectinomycin, and incubated overnight at 37°C. The next day, colony formation was analyzed.

### siRNA knockdown and gene expression analysis

Knockdown of XIAP and NOD1 was performed using HiPerFect Transfection (QIAGEN) of the following siRNAs: siXIAP (AAGGAA-TAAATTGTTCCATGC; QIAGEN), BIRC4_5 (AAGTGCTTTCACTGTGGAGGA; QIAGEN), BIRC4_8 (GGCCGGAATCTTAATATTCGA; QIAGEN), siNOD1 (Hs_CARD4_4, SI00084483; QIAGEN), siRelA (AAGATCAATGGCTA-CACAGGA; QIAGEN), and a non-targeting siRNA (All-Star negative control; QIAGEN).

For gene expression analysis, RT-qPCR analysis was performed. 2 $\mu$g of total RNA was transcribed into cDNA, using the RevertAid First Strand cDNA Synthesis Kit (Thermo Fisher Scientific). qPCR was performed using iQ SYBR Green Supermix (Bio-Rad). The following primers were used: NOD1_fwd: TCCAAAGCCAAACAGAAACTC, NOD1_rev: CAGCATCCA-GATGAACGTG; GAPDH_fwd: GGTATCGTGGAAGGACTCATGAC, GAPDH_rev: ATGCCAGTGAGCTTCCCGTTCAG; and IL-8_fwd: ATGACTTCCAAGCTGGCC GTGGCT, IL-8_rev: TCTCAGCCCTCTTCAAAAACTTCTC.

### Fractionation

Native HeLa cell lysates were prepared by solubilizing cells in Triton X-100 lysis buffer (150 mM NaCl, 50 mM Tris, pH 7.5, and 1% Triton X-100) including proteinase and phosphatase inhibitors (1× Roche complete mini pill, 20 $\mu$M $\beta$-glycerophosphate, 100 $\mu$M sodium-orthovanadate, and 5 mM sodium fluoride) followed by centrifugation for 10 min at 4°C with 21.130$g$ to separate soluble lysate and insoluble fractions. THP-1 cells were lysed in Triton X-100 lysis buffer supplemented with 10% glycerol and 1 mM EDTA followed by centrifugation for 10 min at 4°C with 17.000$g$. The supernatant was used as soluble fraction, and the precipitate was resuspended in lysis buffer with 6 M urea and used as pellet fraction for immunoblotting.

### Immunoblotting

Cells were lysed in SDS buffer (Laemmli), followed by protein denaturation for 5 min at 95°C. Proteins were separated by SDS–PAGE and blotted to 0.2 $\mu$m nitrocellulose membrane, followed by blocking with 0.5% Roche blocking in PBS and incubation with the appropriate antibodies. As primary antibodies anti-$\beta$-actin (sc-47778; Santa Cruz), anti-Flag (F1804; Sigma-Aldrich), anti-GFP (11814460001; Sigma-Aldrich), anti-I$\kappa$B$\alpha$ (#4814; Cell Signaling Technology), anti-RIPK2 (A-10, sc-166765; Santa Cruz), anti-RIPK2 (H-300, sc-22763; Santa Cruz), anti-RIPK2 (#4142; Cell Signaling Technology), anti-RIPK2 pS176

(#4364; Cell Signaling Technology), anti-$\beta$-tubulin (T7816; Sigma-Aldrich), anti-XIAP (E-2, sc-55551; Santa Cruz), and anti-pTyr100 (#9411; Cell Signaling Technology) were used. For detection of primary antibodies antirabbit-HRP (170-6515; Bio-Rad), and antimouse-HRP (170-6516; Bio-Rad) were used. Detection of the signals was performed using Clarity Western ECL Substrate (Bio-Rad), or SuperSignal West Femto maximum sensitivity substrate (Thermo Fisher Scientific) on an automated high sensitivity camera system (Fusion FX; Vilbert Lourmat).

### Immunofluorescence microscopy

For immunofluorescence, the cells were fixed using 4% paraformaldehyde in PBS for 15 min at room temperature, followed by permeabilization with 0.1% Triton X-100 for 5 min. Blocking was performed using 5% fetal calf serum in PBS, after incubation with the primary antibody overnight at 4°C. Primary antibodies used were anti-*Shigella*-LPS (rabbit anti-*S. flexneri* LPS 5a, kindly provided by the laboratory of Philippe Sansonetti, Institute Pasteur), anti-AIF (#4642; Cell Signaling Technology), anti-p65 (sc-8008; Santa Cruz), anti–cleaved-caspase-3 (Asp175, #9661S; Cell Signaling Technology), anti-XIAP (M044-3; MBL), anti-SMN (sc-32313; Santa Cruz), anti-Gemin3 (sc-57007; Santa Cruz), anti-EEA1 (sc-53939; Santa Cruz), anti-LC3$\beta$ (3868; NEB), and anti-Flag M2 (F1804; Sigma-Aldrich). After washing in PBS, incubation with the secondary antibody was performed for 1 h at room temperature. The secondary antibodies used were Alexa 546–conjugated goat antirabbit IgG and Alexa 546–conjugated goat antimouse IgG (Molecular Probes). Coverslips were mounted in Mowiol-Hoechst (Hoechst 33258; Sigma-Aldrich).

For live cell imaging, the cells were cultured in glass-bottom petri dishes (Greiner Bio-one) in FluoroBrite DMEM (Thermo Fisher Scientific), induced with doxycycline (1 $\mu$g/ml) for 16 h and imaged at 37°C in a 5% $CO_2$ atmosphere. Imaging was performed using an automated Leica DMi8 microscope with incubation chamber and the HC PL APO ×63/1.40 oil objective. Images were processed using the Leica LasX software.

### Quantification of RIPosomes

RIPosomes were quantified by eye using blinded processed images. Cells with three or more small bright dots were counted as RIPosome-positive cells. At least 200 cells from one experiment were counted for p65 nuclear translocation, 500 cells for EPEC infection kinetics, and 500 cells for *NOD1* knockdown from two independent experiments.

### NF-$\kappa$B luciferase reporter assays

NF-$\kappa$B activation was measured by a luciferase reporter gene assay. 30,000 cells were plated per well of a 96-well plate and transiently transfected with X-tremeGENE9. 8.6 ng of $\beta$-galactosidase plasmid, 13 ng of the luciferase reporter plasmid, and 10, 5, or 1 ng of EGFP-RIPK2 expression plasmids were transfected into the cells, using a total DNA amount of 51 ng plasmid per well. After incubation, the cells were lysed, and luciferase activity was quantified on a luminometer. $\beta$-galactosidase activity was measured by ONPG

assay. Luciferase activity was normalized to the *β*-galactosidase activity.

## ELISA

IL-8 (CXCL8) was measured in cell culture supernatants using a Duoset (DY208; Bio-Techne) according to the manufacturer's instructions.

## Statistical analysis

Data were analyzed and plotted using Microsoft Excel and GraphPad Prism 7.0. A *t* test was used to determine statistical significance.

# Supplementary Information

# Acknowledgements

We thank Mariana Mohr for help in generating the EGFP-RIPK2 line and Sarah Bauer for assistance in RIPosome quantification. This work was supported by the German Research Foundation grant KU 1945/4-1 to TAK and the German Academic Exchange Service (DAAD) project PPP 57445802. This work was also made possible through Victorian State Government Operational In-frastructure Support and Australian Government NHMRC IRIISS (9000220).

## Author Contributions

K Ellwanger: data curation, formal analysis, supervision, validation, investigation, visualization, methodology, and writing—review and editing.
S Briese: data curation, formal analysis, visualization, methodology, and writing—review and editing.
C Arnold: data curation, formal analysis, visualization, methodology, and writing—original draft.
I Kienes: data curation, formal analysis, and methodology.
V Heim: data curation, formal analysis, and methodology.
U Nachbur: supervision, funding acquisition, and project administration.
K Thomas: conceptualization, supervision, funding acquisition, project administration, and writing—review and editing.

## Conflict of Interest Statement

The authors declare that they have no conflict of interest.

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
