## [Reviewer comments · Life Science Alliance]

Life Science Alliance

XIAP controls RIPK2 signaling by preventing its deposition in speck-like structures

Kornelia Ellwanger, Selina Briese, Christine Arnold, Ioannis Kienes, Valentin Heim, Ueli Nachbur, and Kufer Thomas

DOI: <https://doi.org/10.26508/lsa.201900346>

Corresponding author(s): Kornelia Ellwanger, Institute of Nutritional Medicine, University Hohenheim

Review Timeline:

Submission Date:	2019-02-14
Editorial Decision:	2019-03-11
Revision Received:	2019-07-04
Editorial Decision:	2019-07-15
Revision Received:	2019-07-15
Accepted:	2019-07-16

Scientific Editor: Andrea Leibfried

Transaction Report:

March 11, 2019

Re: Life Science Alliance manuscript #LSA-2019-00346-T

Dr. Kornelia Ellwanger
Institute of Nutritional Medicine, University Hohenheim
Immunology
Fruwirthstr. 12
Stuttgart 70599
Germany

Dear Dr. Ellwanger,

Thank you for submitting your manuscript entitled "XIAP controls RIPK2 signaling by preventing its deposition in speck-like structures" to Life Science Alliance. The manuscript was assessed by expert reviewers, whose comments are appended to this letter.

As you will see, the reviewers find your work interesting, but voice overlapping concerns regarding the support provided for the conclusions put forward. They both request extension to other cell lines and more functional insight. They also raise some additional technical concerns.

We agree with the reviewers that your work is interesting, and we would thus like to invite you to submit a revised manuscript that addresses the concerns raised. We realize that addressing these will require a lot of effort and time, and we'd be happy to extend the revision time. We do not expect you to provide full insight into Riposome function, but at least some data to support a functional role should get provided. Please note that we'd need strong support from the reviewers on a revised version, so please consider your options carefully and let us know in case you'd rather seek publication elsewhere.

Please note that papers are generally considered through only one revision cycle.

Thank you for this interesting contribution to Life Science Alliance. We are looking forward to receiving your revised manuscript.

Sincerely,

B. MANUSCRIPT ORGANIZATION AND FORMATTING:

Reviewer #1 (Comments to the Authors (Required)):

Ellwanger et al: XIAP controls RIPK2 signaling by preventing its deposition in speck-like structures

This is interesting work of Ellwanger and colleagues to investigate the molecular signalling mechanisms of RIPK2. They find that RIPK2 redistributes to speck-like structures after bacterial infections, RIPK2 overexpression, XIAP inhibition as well as RIPK2 inhibition. Assembly in speck-like structures is independent of NFkB activation and happens only after the signalling events that promote cytokine production. Furthermore the authors identify two novel binders to RIPK2, Erlin and 14-3-3 proteins that associate specifically with RIPK2 in the active, respectively inactive state.

The observation that RIPK2 assembles in higher order structures is interesting and has provoked interest in the scientific community. Here the authors show a rather surprising timing of these speck-assembly, which happens significantly after the main signalling events such as NFkB activation. The relevance of this RIPosome assembly remains therefore unclear.

The data presented is generally supportive of the claims made by the authors. Critically, the authors use overexpression as well as endogenous RIPK2 in HeLa cells to investigate RIPK2 signaling. Overexpression is notoriously problematic as induction of members of the NOD-RIPK2 signaling pathway results in auto activation of the pathway. However this issue has been addressed and discussed and at least in this system, overexpression seem to mimic the endogenous systems. However the study is currently restricted to HeLa cells and would benefit from additional, more relevant cells such as primary macrophages and an extension to NOD2.

Conceptually, I find it difficult to understand why RIPK2 assembles into speck-like structures after the main signalling events, raising the question of its relevance. Accumulation of signalling molecules after stimulation is seen in many innate immune pathways, and contributes to enhance signalling efficacy. In this case, however, even stimuli that negatively regulate RIPK2 signaling, such as RIPK2 inhibition or antagonism of XIAP result in speck-like structures of RIPK2. The timing of the redistribution could suggest a mechanism that this is part of a recycling or a shut down mechanism. The authors could therefore test whether speck-associated RIPK2 is being modified by K48-linked ubiquitin chains, respectively whether inhibition of the proteasome results in enhanced speck-association of RIPK2. Potentially, the MS dataset used for Figure 7 could be interrogated for PTMs of RIPK2 and associated ubiquitins.

If XIAP has a central role in controlling RIPK2 speck formation, what happens in XIAP knock-out cells? Previous studies showed that XIAP deficient cells have normal RIPK2 levels (for instance Damgaard et al, Mol Cell 2012), which is contradictory to the results here. In light of the interest of IAP antagonists in the clinic, the authors could also test SMAC mimetic compounds on various cell lines.

It is also not clear whether RIPosome formation is dependent on a transcriptional response. The authors show that RIPosome formation is independent on (canonical) NFkB activation by knock-down of RelA, but non-canonical NFkB, respectively MAPK dependent activation of AP-1 could still be required.

The work on Erlin/ 14-3-3 seems premature and does not quite fit in the context of the paper. The physiological relevance of these interactions should be investigated if this part is to remain in the manuscript.

Further comments

Fig 1: In panel A and B, the authors should include uninfected controls, particularly as an overexpression system is used. Legend to panel B makes no sense. Legend to panel E includes description of panel F. Was quantification of the staining (panel E) only performed once (no error bars)? For quantification, all experiments should be considered.

Fig 5: In panel B, efficacy of XIAP knock-down should be shown. In panel D, according to the figure legend, the cells are 'infected with *S. flexneri* M90T or BS176...'. Which one was actually used? In panel E, levels of XIAP, and possibly cIAPs should be shown to show efficacy of SMAC constructs.

Fig S6: The data using the RIPK2 inhibitors Gefitinib/GSK583 is intriguing. The specks with the GSK compound are clearly different from Gefitinib treated cells.

Minor comments:

Abstract, line 9: The IAP (not cIAP) protein XIAP...

Page 6, 2nd paragraph, line 6 refers to figure 3B, not 4B

Fig 4: The formatting of this legend is different from the other legends.

Fig 5E: The order of SMAC constructs used should be the same for the micrographs and the Western blot (too confusing at the moment)

Reviewer #2 (Comments to the Authors (Required)):

The manuscript by Ellwanger et al. describes the formation of two distinct RIPK2 complexes upon *Shigella* infection - a soluble complex that presumably activates NF- κ B and an "Speck-like" Triton-insoluble complex that functions in an unknown manner. The authors find that these states are regulated both by XIAP binding and by tyrosine phosphorylation of RIPK2. The authors further show that RIPK2 binds to the 14-3-3 family of proteins and to two proteins implicated in ERAD. I have mixed feelings about the work. On the positive side, there are clearly two distinct forms of RIPK2 complexes. This has only been studied in a limited manner and the author's biochemistry and PTM breakdown is generally convincing. On the more critical side, the authors never find the function of the "Speck-like" complexes. It appears observational - there aren't any studies implicating these complexes in bacteriocidal or bacteriostatic functions, there aren't any studies that rule out that these complexes are aggresomes or sequestered signaling complexes and there aren't any studies that study these complexes in any immunologic context. Coupled with the fact that the last figure - mass spec analysis of RIPK2 binding partners - seems tacked on without any additional study, I find the manuscript missing biochemical mechanism and seeming disjointed and preliminary. My major and minor comments are as follows:

Major

1. The argument seems to be that XIAP and pY474 maintain a soluble form of the NOD-RIPK2

complex that exerts signaling function while the "speck-like" complex exerts a separate function. The authors never find a function of the "speck-like" complex and never rule out that this is simply a sequestome that decreases NF- κ B function.

2. The cell line generated isn't clearly described and may be subject to artifact. GFP is roughly a 26 kD protein and increases the size of RIPK2 by 50%. The manuscript states that they are replacing at endogenous levels, but endogenous levels are never shown (all blots are > 70 kD). It is unclear that WT RIPK2 isn't functional in these cells and it is unclear that GFP-RIPK2 completely replicates WT signaling. It would be much better to CRISPR knockout endogenous RIPK2 before replacement.

3. Along these same lines, only a single cell line is chosen - dox inducible GFP RIPK2 HeLa cells. Does the finding replicate across cell lines and into the myeloid lineage with actual endogenous RIPK2?

4. A single RNAi is insufficient. CRISPR is much better, but at the very least >2 RNAis need to be used in the experimentation.

5. Are the authors sure that the NOD1 and NOD2 mutants used are gain-of-function? Aside from the initial Blood paper showing NF- κ B luciferase assays, the EOS NOD2 mutants have not been universally shown to be gain of function.

6. The results in Figure 6 don't necessarily follow the model. S176A and S176E both induce Specks at earlier time points - they show increased NF- κ B activity, though. Why is this? My read of the manuscript is that the soluble fraction supports NF- κ B activation.

7. Figure 7 is tacked on and doesn't fit the manuscript at all. Do the 14-3-3 proteins or Erlin regulate Speck formation? Much of the co-IP data isn't convincing, and there aren't any functional assays on these findings.

8. The findings in Supplemental Figure 6 are interesting. The Goncharov Molecular Cell paper showed that the GSK compound caused loss of XIAP binding while Gefitinib did not with the argument being that the kinase domain functions solely to bind XIAP. Figure S6 shows something completely different though. GSK shows a filamentous RIPK2 complex while Gefitinib shows the more speck-like formation. There needs to be an explanation of the disparate results with the Molecular Cell paper.

Minor

1. It isn't clear to this reviewer what the TNF or RelA studies add.

2. Figure 1D doesn't match the kinetics of p65 nuclear translocation with the graph. It looks to this reviewer that nuclear p65 is roughly concordant with speck formation.

3. In the introduction, I wouldn't say that RIPK2 affects T cell signaling. This is controversial at best and probably in the minority of thought.

4. Given the inflammasome nomenclature, calling these complexes "Specks" is problematic.

In summary, I feel that there are the beginnings of a strong manuscript here. The two complex phenomena is likely real and deserves study. The function of the speck complex isn't delineated,

though, and this really needs to be done. Additionally, there are numerous experimental issues as outlined above. While some of the data is interesting, a significant amount of additional mechanism and experimental controls need to be done to increase enthusiasm.

Reviewer #1:

The observation that RIPK2 assembles in higher order structures is interesting and has provoked interest in the scientific community. Here the authors show a rather surprising timing of these speck-assembly, which happens significantly after the main signalling events such as NF κ B activation. The relevance of this RIPosome assembly remains therefore unclear.

We agree with the reviewer that our data unfortunately could not unravel a clear function for RIPosomes per se. In the revised version, we provide evidence for the requirement of RIPK2 ubiquitination for RIPosome formation by the use of RIPK2 lysine to arginine mutants (new panel A-D in figure 5). Moreover, we expanded our analyses on other cell types (see below). Concerning the unexpected timing of RIPosome formation and NF- κ B activation, we corroborated these data by adding more experiments and by adding measurements of IL-8 mRNA expression (shown in the revised figure 1 panel H and I). However, we would like to emphasize that our novel observations and generated tools provide the means to study novel aspects of RIPK2 activation in the community.

The data presented is generally supportive of the claims made by the authors. Critically, the authors use overexpression as well as endogenous RIPK2 in HeLa cells to investigate RIPK2 signaling. Overexpression is notoriously problematic as induction of members of the NOD-RIPK2 signaling pathway results in auto activation of the pathway. However this issue has been addressed and discussed and at least in this system, overexpression seem to mimic the endogenous systems. However the study is currently restricted to HeLa cells and would benefit from additional, more relevant cells such as primary macrophages and an extension to NOD2.

We used the HeLa Flp-In cell lines as cell-based system to further analyse RIPK2 activation in a controlled setting. We fully agree with the reviewer that the use of additional cell types would support our claims. We thus used human and mouse myeloid cell lines and show that overexpression of EGFP-RIPK2 but not of RIPK2 Y474F induces RIPosomes in these cells (shown in the revised figure S6, new panel F). Moreover, we show that endogenous RIPK2 from human THP-1 cells also forms detergent insoluble complexes upon activation of the cells with NOD2 ligands and treatment with XIAP inhibitors (shown in the revised figure 4, new panel E).

Conceptually, I find it difficult to understand why RIPK2 assembles into speck-like structures after the main signalling events, raising the question of its relevance. Accumulation of signalling molecules after stimulation is seen in many innate immune pathways, and contributes to enhance signalling efficacy. In this case, however, even stimuli that negatively regulate RIPK2 signaling, such as RIPK2 inhibition or antagonism of XIAP result in speck-like structures of RIPK2. The timing of the redistribution could suggest a mechanism that this is part of a recycling or a shut down mechanism.

The authors could therefore test whether speck-associated RIPK2 is being modified by K48-linked ubiquitin chains, respectively whether inhibition of the proteasome results in enhanced speck-association of RIPK2. Potentially, the MS dataset used for figure 7 could be interrogated for PTMs of RIPK2 and associated ubiquitins.

To address if RIPK2 is K48 ubiquitinated we performed immunoprecipitations of EGFP-RIPK2 upon *Shigella* infection and probed with a K48 ubiquitin specific antibody. This suggested that RIPK2 gets K48-ubiquitinated upon infection and that this event coincides with upshift of RIPK2 (see data below). However, as this experimental approach is not suited to directly prove K48 ubiquitination of RIPK2, we decided to not include these data in the manuscript.

Immunoblot analysis of anti-GFP immunoprecipitates of HeLa EGFP-RIPK2 cells at different timepoints after infection with *S. flexneri* M90T. Probing for K48 Ubiquitin, RIPK2 and β -actin as loading control. Immunoprecipiated proteins are shown in the left panel (IP) and input fraction (total cell lysate, TCL) in the right panel.

Formation of complexes of RIP-kinases by negative regulators, i.e. XIAP was also observed for RIPK3 (Yabal et al. CellReports 2014) and thus is not so unexpected.

As suggested by the reviewer, we tested the effect of proteasome inhibition using MG132. Treating the EGFP-RIPK2 HeLa cells with MG132 induced sterile RIPosome formation starting at 4 h as evidenced by immunofluorescence and immunoblot analysis. This novel data is shown in the revised figure S7 panel D. This supports the idea that RIPosomes may be a platform to degrade or shut down RIPK2 signaling. However, in our view, this does not formally exclude that RIPosomes have additional signaling functions that remain to be identified.

If XIAP has a central role in controlling RIPK2 speck formation, what happens in XIAP knock-out cells? Previous studies showed that XIAP deficient cells have normal RIPK2 levels (for instance Damgaard et al, Mol Cell 2012), which is contradictory to the results here. In light of the interest of IAP antagonists in the clinic, the authors could also test SMAC mimetic compounds on various cell lines.

In our view, the observation by Damgaard et al. goes well in line with the results we obtained using XIAP knock-down. In our experiments, the overall protein levels of RIPK2 were not changed (see Fig. 4C). This might indicate that XIAP deficiency is not sufficient to induce

degradation of RIPK2. To address this in more detail we used the SMAC mimetic compound A on THP-1 cells. In THP-1 cells, we observed that NOD2 activation alone had virtually no effect on RIPK2 complex formation but co-stimulation with a NOD2 agonist and the XIAP-inhibitory compound A induced RIPK2 complex formation (i.e. electro mobility shift and detergent insolubility). These novel data are depicted in the revised figure 4E. The differential requirement of NOD stimulation between THP-1 and the HeLa EGFP-RIPK2 cells for formation of RIPOsomes upon XIAP targeting can be explained by the fact that our RIPK2 overexpressing HeLa cell line showed a certain degree of auto-activation (see figure 1B) which might be necessary to drive RIPOsome formation upon XIAP depletion but is not sufficient to induce full RIPK2 activation without stimulation of NOD1/2.

It is also not clear whether RIPOsome formation is dependent on a transcriptional response. The authors show that RIPOsome formation is independent on (canonical) NF κ B activation by knock-down of RelA, but non-canonical NF κ B, respectively MAPK dependent activation of AP-1 could still be required.

We agree that we cannot formally rule out that RIPOsome formation might be induced by a yet to be identified signaling pathway. However, the fact that XIAP silencing induces RIPOsomes and our data on NOD-dependency and NF- κ B-independency of this effect make this not very likely. Still, we considered the reviewers alternative interpretation in the discussion and edited the main text to cover this aspect more accurately.

The work on Erlin/ 14-3-3 seems premature and does not quite fit in the context of the paper. The physiological relevance of these interactions should be investigated if this part is to remain in the manuscript.

As also suggested by reviewer #2, we decided to exclude these data from the revised manuscript.

Further comments

Fig 1: In panel A and B, the authors should include uninfected controls, particularly as an overexpression system is used. Legend to panel B makes no sense. Legend to panel E includes description of panel F. Was quantification of the staining (panel E) only performed once (no error bars)? For quantification, all experiments should be considered.

In former panel A and B of figure 1, which is now the revised figure 1B+C we have used BS176 as control. We would like to argue that this is an even better control than using uninfected cells. BS176 are *Shigella flexneri* devoid of the invasion plasmid, thus allowing us to control for changes brought about by bacterial metabolism and the procedure of infection. We apologize for the errors in the figure legend that we now corrected. We replaced panel E, now showing quantification of two independent experiments including S.D. (now panel H in the revised figure 1).

To better visualize p65 activation kinetics, we also changed the former panel D of figure 1 and now show results from an independent experiment, where p65 sub-cellular localization was better visible in the immunofluorescence staining (panel G of the revised figure 1). Furthermore, to support our conclusion on RIPosome formation timing we conducted qPCR analysis to measure IL-8 mRNA expression (see also comment from reviewer #2). The novel qPCR data are shown in the revised figure 1, panel I.

Fig 5: In panel B, efficacy of XIAP knock-down should be shown. In panel D, according to the figure legend, the cells are 'infected with *S. flexneri* M90T or BS176...'. Which one was actually used? In panel E, levels of XIAP, and possibly cIAPs should be shown to show efficacy of SMAC constructs.

The efficiency of XIAP knock-down was evaluated by immunoblot and is shown in the revised figure 4 in panel C (corresponds to the former figure 5). Cells used for infection and fluorescence microscopy (shown in the revised figure 4 panel B) were derived from the same experiment as the cells used for infection and lysis for immunoblot (shown in the revised figure 4 panel C).

Furthermore, to rule out off-target-effects of siRNA (see comment from reviewer #2), we applied two additional validated siRNAs to silence XIAP and obtained similar results concerning silencing efficiency and RIPosome induction. These data are shown in the revised figure S5 panels A+B.

We apologize for the error in the figure legend. The cells were infected only with *S. flexneri* M90T, this information was corrected in the revised figure. Following the reviewers suggestion, a blot showing XIAP probing was added to the immunoblot for the SMAC constructs. This is now shown in the revised figure S5 panel D.

Fig S6: The data using the RIPK2 inhibitors Gefitinib/GSK583 is intriguing. The specks with the GSK compound are clearly different from Gefitinib treated cells.

In the revised version of figure S7B we added immunoblots for XIAP and β -actin as loading control. This panel now illustrates that the two inhibitors also differentially affect levels of XIAP and S176 phosphorylation of RIPK2 in the course of *S. flexneri* M90T infection. We conclude that this is due to the differential mode of action of the two compounds with GSK583 inhibiting binding of XIAP to RIPK2 and Gefitinib not affecting XIAP:RIPK2 interference (as suggested by Goncharov et al, 2018). We added further discussion of these data in the manuscript.

Minor comments:

Abstract, line 9: The IAP (not cIAP) protein XIAP...

Page 6, 2nd paragraph, line 6 refers to figure 3B, not 4B

Fig 4: The formatting of this legend is different from the other legends.

Fig 5E: The order of SMAC constructs used should be the same for the micrographs and the Western blot (too confusing at the moment)

We thank the reviewer for the accurate reading of the manuscript. We apologize for these errors. All above indicated errors have been rectified in the revised version.

Reviewer #2:

Major points

1. The argument seems to be that XIAP and pY474 maintain a soluble form of the NOD-RIPK2 complex that exerts signaling function while the "speck-like" complex exerts a separate function. The authors never find a function of the "speck-like" complex and never rule out that this is simply a sequestome that decreases NF- κ B function.

We agree with the reviewer that we were unfortunately not able to unravel a defined function of the RIPOsomes so far. In line with the reviewer's suggestion, we think that RIPOsomes might be compartments to dampen RIPK2 signaling and eventual for protein degradation. We conducted novel experiments and show that proteasome inhibition by MG132 induces RIPOsome formation in the HeLa EGFP-RIPK2 cells (shown in the revised figure S7 panel D). Our data thus do not exclude the possibility that RIPOsomes are sequestome-associated complexes, although we think that these complexes are different from classical sequestomes and might have additional functions in the cell. Most importantly, the bacteria-induced formation of these complex is a suited reporter for changes in the biochemical properties of RIPK2. This interpretation is included in the revised main text.

2. The cell line generated isn't clearly described and may be subject to artifact. GFP is roughly a 26 kD protein and increases the size of RIPK2 by 50%. The manuscript states that they are replacing at endogenous levels, but endogenous levels are never shown (all blots are > 70 kD). It is unclear that WT RIPK2 isn't functional in these cells and it is unclear that GFP-RIPK2 completely replicates WT signaling. It would be much better to CRISPR knockout endogenous RIPK2 before replacement.

We edited the figures and the manuscript and hope that this helps to clarify this point. In our HeLa Flp-In cell culture model we did not replace endogenous RIPK2. Instead, we facilitate the doxycycline inducible overexpression of EGFP, EGFP-RIPK2 or EGFP-RIPK2 with specific point mutations. In the revised figure 1 panel B we show that by doxycycline induced overexpression of EGFP-RIPK2 we can increase basal and *Shigella flexneri* induced NF- κ B responses, as we would expect.

We added molecular weight markers for the immunoblots in figure 1 to emphasize the molecular weight of the detected proteins. In figure 1 panel C+D we show overexpressed EGFP-RIPK2, which runs at ~90 kDa. In figure 1 panel E we detected endogenous RIPK2 in HeLa cells, which runs at about ~60 kDa, as expected. We added new data on endogenous RIPK2 in THP-cells, which also runs at ~60 kDa (figure 4 panel E).

We principally agree that knock-out of endogenous RIPK2 from these cells and replacement by the EGFP-tagged variant would be informative. However, these experiments are beyond the scope of the possibilities of a revision. Moreover, CRISPR targeting has intrinsic problems (see Kosicki, Tomberg and Bradley, Nat.Biotechnology 2018). To add to the reviewer's query, we provide novel data on endogenous RIPK2 also in myeloid cells (revised

figure 4 panel E) supporting our main conclusions of RIPosome formation and the role of XIAP.

3. Along these same lines, only a single cell line is chosen - dox inducible GFP RIPK2 HeLa cells. Does the finding replicate across cell lines and into the myeloid lineage with actual endogenous RIPK2?

We already provided some evidence that also endogenous RIPK2 from HeLa cells forms insoluble aggregates upon activation (see revised figure 1 panel E). However, we fully agree with the reviewer that the use of additional cell types would substantiate the general significance of our findings. We thus used myeloid cells from human and mouse and found that overexpression of the WT but not the Y474F form of RIPK2 induced RIPosomes also in these cells (this is shown in the revised figure S6 new panel F). This supports that RIPosome formation is not cell line specific. Following the reviewers suggestion, we also used THP-1 cells to address if endogenous RIPK2 also forms RIPosomes. Treatment of these cells with NOD2 agonist and SMAC mimetics led to a strong upshift and detergent insolubility of endogenous RIPK2 from these cells. This novel data is shown in the revised figure 4 panel E. The differential requirement of NOD stimulation between THP-1 and the HeLa EGFP-RIPK2 cells for formation of RIPosomes upon XIAP targeting can be explained by the fact that our RIPK2 overexpressing HeLa cell line showed a certain degree of auto-activation (see figure 1B) which might be necessary to drive RIPosome formation upon XIAP depletion but is not sufficient to induce full RIPK2 activation without stimulation of NOD1/2 (see also comment by reviewer #1).

4. A single RNAi is insufficient. CRISPR is much better, but at the very least >2 RNAis need to be used in the experimentation.

Following the reviewers suggestion, we conducted experiments with two novel siRNA for XIAP along with the duplex used in the initial experiments. All three siRNA led to similar results in terms of XIAP knockdown efficiency and RIPosome formation. This novel data is shown in the revised supplementary figure S5 panels A+B.

5. Are the authors sure that the NOD1 and NOD2 mutants used are gain-of-function? Aside from the initial Blood paper showing NF-kB luciferase assays, the EOS NOD2 mutants have not been universally shown to be gain of function.

We agree with the reviewer that it is difficult to provide a clear statement on the genetics of these mutations. We thus avoided the use of the term “gain-of-function”. However, what others and we showed is that the used NOD1 and NOD2 mutants have different signaling activity in the in vitro test used here.

6. The results in figure 6 don't necessarily follow the model. S176A and S176E both induce Specks at earlier time points - they show increased NF- κ B activity, though. Why is this? My read of the manuscript is that the soluble fraction supports NF- κ B activation.

We respectfully disagree with the reviewer concerning the speck formation of these mutants. When looking at the data from the former figure 6 (now figure 5 panels E-I of the revised manuscript), the S176A mutation induced more rapidly the formation of specks compared to WT whereas the phosphomimetic S176E showed slightly delayed RIPosome formation (revised figure 5 panel F). The reviewer, however well spotted the discrepancy between this and a model where soluble RIPK2 is the active form. However, dephosphorylation at S176 likely is a prerequisite for RIPK2 signaling, explaining the higher NF- κ B activation of the S176A mutant and at the same time the enhanced RIPosome formation, which is a read out of RIPK2 activation as shown by *Shigella* infection. The S176E accordingly would be trapped in a more inactive formation, thus induced less NF- κ B.

7. Figure 7 is tacked on and doesn't fit the manuscript at all. Do the 14-3-3 proteins or Erlin regulate Speck formation? Much of the co-IP data isn't convincing, and there aren't any functional assays on these findings.

As suggested also by reviewer #1, we decided to exclude this data from the revised manuscript.

8. The findings in Supplemental figure 6 are interesting. The Goncharov Molecular Cell paper showed that the GSK compound caused loss of XIAP binding while Gefitinib did not with the argument being that the kinase domain functions solely to bind XIAP. Figure S6 shows something completely different though. GSK shows a filamentous RIPK2 complex while Gefitinib shows the more speck-like formation. There needs to be an explanation of the disparate results with the Molecular Cell paper.

In the revised version of figure S7 in panel B we added immunoblots for XIAP and β -actin as loading control. Besides showing the formation of higher molecular weight signals at later timepoints of infection for both inhibitors this figure also shows that Gefitinib and GSK583 differentially affect XIAP and pSer176 levels. XIAP levels were preserved and S176 phosphorylation increased in the course of *S. flexneri* M90T infection when cells were pretreated with GSK583, whereas Gefitinib led to reduced XIAP levels and pS176 (figure S7 panel B). We conclude that this is due to the differential mode of action of the two compounds with GSK583 inhibiting binding of XIAP to RIPK2 and Gefitinib not affecting XIAP:RIPK2 interference (as suggested by the reviewed based on Goncharov et al, 2018). The effect that GSK583 induced RIPosomes is in line with Goncharov et al., at present we unfortunately can not provide explanations for the unexpected finding that Gefitinib induced RIPosomes as well

as for the different quality of the structures. We also expanded the discussion section of these data in the manuscript.

Minor points

1. It isn't clear to this reviewer what the TNF or RelA studies add.

We are sorry if this was not described well enough. The intent was to exclude that a canonical NF- κ B driven event or NF- κ B signaling itself leads to RIPosome formation. We edited the text for better clarity.

2. figure 1D doesn't match the kinetics of p65 nuclear translocation with the graph. It looks to this reviewer that nuclear p65 is roughly concordant with speck formation.

To better visualize p65 activation kinetics, we changed the former panel D of figure 1 and now show results from an independent experiment, where p65 sub-cellular localization was better visible (see revised figure 1 panel G). In order to substantiate this data, we further provide quantification showing S.D. derived from independent experiments (revised figure 1 panel H) and also performed qPCR to quantify *IL-8* mRNA (shown in the revised figure 1 panel I). This clearly supports that both p65 nuclear translocation and *IL-8* mRNA expression occurred prior to RIPK2 aggregation.

3. In the introduction, I wouldn't say that RIPK2 affects T cell signaling. This is controversial at best and probably in the minority of thought.

We agree with the reviewer and toned down this statement.

4. Given the inflammasome nomenclature, calling these complexes "Specks" is problematic.

We principally agree with the reviewer that the term specks is associated with inflammasomes. However, we never refer to these structures as specks and only use the term "speck-like structures" in the title. Given the similarity to ASC and the presence of a conserved phosphorylation site that is involved in complex formation of both proteins, we decided to keep this wording.

July 15, 2019

RE: Life Science Alliance Manuscript #LSA-2019-00346-TR

Dr. Kornelia Ellwanger
Institute of Nutritional Medicine, University Hohenheim
Immunology
Fruwirthstr. 12
Stuttgart 70599
Germany

Dear Dr. Ellwanger,

Thank you for submitting your revised manuscript entitled "XIAP controls RIPK2 signaling by preventing its deposition in speck-like structures". One of the original reviewers re-assessed your work and appreciates the introduced changes, and we would thus be happy to publish your paper in Life Science Alliance pending final revisions necessary to meet our formatting guidelines:

- please address the remaining concern of the reviewer by text changes
- please provide the manuscript text in word docx file format
- please upload the S figure files as individual files
- please add callouts in the manuscript text for Fig S1A,B, Fig S3B,D,E, Fig S5A,B
- some figures are too tightly cropped (bottom or right hand side), please amend
- one of your co-authors is not listed as an author on the revised manuscript anymore; please explain and provide written confirmation from all authors (incl the co-author not listed anymore) that this change in authorship is appropriate

A. FINAL FILES:

-- High-resolution figure, supplementary figure and video files uploaded as individual files: See our detailed guidelines for preparing your production-ready images, <http://www.life-science->

alliance.org/authors

B. MANUSCRIPT ORGANIZATION AND FORMATTING:

Sincerely,

Andrea Leibfried, PhD
Executive Editor
Life Science Alliance
Meyerhofstr. 1
69117 Heidelberg, Germany
t +49 6221 8891 502
e a.leibfried@life-science-alliance.org

Reviewer #2 (Comments to the Authors (Required)):

The authors have satisfactorily addressed my concerns - although I do wish a function for these complexes could be found, I think its outside the scope of publication.

One point in regards to Crispr v. RNAi. The author cites a Nature Biotechnology manuscript which outlines potential off-target genetic issues with Crispr. Given Nature Publishing Group's spotty history with Crispr papers (some since retracted - Nature Methods 14:547), and given RNAi's known IFN effects which is especially problematic in innate immune signaling, I might not discount using Crispr in these cases.

At any rate, these are minor points and I am supportive of the manuscript.

July 16, 2019

RE: Life Science Alliance Manuscript #LSA-2019-00346-TRR

Dr. Kornelia Ellwanger
Institute of Nutritional Medicine, University Hohenheim
Immunology
Fruwirthstr. 12
Stuttgart 70599
Germany

Dear Dr. Ellwanger,

Thank you for submitting your Research Article entitled "XIAP controls RIPK2 signaling by preventing its deposition in speck-like structures". It is a pleasure to let you know that your manuscript is now accepted for publication in Life Science Alliance. Congratulations on this interesting work.

DISTRIBUTION OF MATERIALS:

Again, congratulations on a very nice paper. I hope you found the review process to be constructive and are pleased with how the manuscript was handled editorially. We look forward to future exciting submissions from your lab.

Sincerely,

Andrea Leibfried, PhD
Executive Editor
Life Science Alliance
Meyerohofstr. 1
69117 Heidelberg, Germany
t +49 6221 8891 502
e a.leibfried@life-science-alliance.org
www.life-science-alliance.org